# A New Potential Therapeutic Target for Cancer in Ubiquitin-Like Proteins—UBL3

**DOI:** 10.3390/ijms24021231

**Published:** 2023-01-08

**Authors:** Hengsen Zhang, Bin Chen, A. S. M. Waliullah, Shuhei Aramaki, Yashuang Ping, Yusuke Takanashi, Chi Zhang, Qing Zhai, Jing Yan, Soho Oyama, Tomoaki Kahyo, Mitsutoshi Setou

**Affiliations:** 1Department of Cellular and Molecular Anatomy, Hamamatsu University School of Medicine, 1-20-1 Handayama, Higashi-Ku, Hamamatsu, Shizuoka 431-3192, Japan; 2Department of Radiation Oncology, Hamamatsu University School of Medicine, 1-20-1 Handayama, Higashi-Ku, Hamamatsu, Shizuoka 431-3192, Japan; 3First Department of Surgery, Hamamatsu University School of Medicine, 1-20-1 Handayama, Higashi-Ku, Hamamatsu, Shizuoka 431-3192, Japan; 4Department of Systems Molecular Anatomy, Institute for Medical Photonics Research, Preeminent Medical Photonics, Education & Research Center, 1-20-1 Handayama, Higashi-Ku, Hamamatsu, Shizuoka 431-3192, Japan; 5International Mass Imaging Center, Hamamatsu University School of Medicine, 1-20-1 Handayama, Higashi-Ku, Hamamatsu, Shizuoka 431-3192, Japan

**Keywords:** post-translational modification, ubiquitin like, UBL3, cancer

## Abstract

Ubiquitin-like proteins (Ubls) are involved in a variety of biological processes through the modification of proteins. Dysregulation of Ubl modifications is associated with various diseases, especially cancer. Ubiquitin-like protein 3 (UBL3), a type of Ubl, was revealed to be a key factor in the process of small extracellular vesicle (sEV) protein sorting and major histocompatibility complex class II ubiquitination. A variety of sEV proteins that affects cancer properties has been found to interact with UBL3. An increasing number of studies has implied that UBL3 expression affects cancer cell growth and cancer prognosis. In this review, we provide an overview of the relationship between various Ubls and cancers. We mainly introduce UBL3 and its functions and summarize the current findings of UBL3 and examine its potential as a therapeutic target in cancers.

## 1. Introduction

Post-translational modification (PTM) of proteins is the addition of modifying groups to specific amino acids in a conjugate manner after protein biosynthesis to precisely modulate protein properties and optimize cellular processes. Ubiquitin (Ub) and ubiquitin-like proteins (Ubls) have been identified as the third-most-common type of PTMs after phosphorylation and glycosylation [1]. They play a critical role in protein homeostasis by regulating a large number of biological processes, such as transcription, DNA repair, signal transduction, autophagy, cell-cycle control, and protein stability [2,3]. Ubls are a family of small proteins, structurally similar to Ub, and most Ubls have been identified to be involved in PTMs of cellular proteins. Ub and Ubls have now been shown to be involved in the pathological processes of a variety of diseases, including various types of cancers [4].

Ubiquitin-like 3 (UBL3), a highly conserved Ubl called membrane-anchored ubiquitin-fold protein (MUB), was identified in *Arabidopsis* [5]. With the deep investigation of UBL3, the functions of UBL3 have been gradually revealed. UBL3 has been reported to interact non-covalently with *Arabidopsis* group VI ubiquitin-conjugating E2 enzymes in vitro, and prevent E2 activation and E2 ubiquitin formation via anchoring at the plasma membrane [6]. In 2018, it was reported that UBL3 can act as a protein post-translational modifier to sort specific proteins, including RAS oncoprotein, into small extracellular vesicles (sEVs) [7]. sEVs are nanoscale bimolecular lipid vesicles secreted from multivesicular bodies (MVBs) in cells. The exosome is an sEV, defined by sizes of 30–150 nm in diameter. The roles of sEVs have been discussed in the context of progression and metastasis in various cancers [8,9,10]. sEVs are capable of transporting molecular cargo from one cell to another through vascular vessels and intercellular spaces. As a form of intercellular communication, the sEV system regulates the function and phenotype of recipient cells [11].

UBL3 was also reported to regulate the membrane-associated RING-CH (MARCH) E3 ligases that promote the ubiquitination of major histocompatibility complex class II (MHC II) and cluster of differentiation 86 (CD86) and affect dendritic cell number and function [12]. This revealed an essential role of UBL3 in the immune system. Dendritic cell presentation of tumor antigens to CD4+ and CD8+ T-lymphocytes is critical for tumor immunity, and CD86 has also been found to be a biomarker for a variety of tumor immune-related prognoses [13,14,15]. UBL3 has the potential to be a key target for cancer therapy.

In this review, we provide an overview of the research on Ubls in cancer. Particularly, we highlight UBL3’s function, its relationship with various cancers, and the potential in cancer therapy.

## 2. Ub and Ubls

Ub is a highly conserved protein consisting of 76 amino acids and is expressed in nearly all eukaryotic cells [16]. Ub can regulate protein homeostasis by participating in protein degradation through covalent bonding with target proteins (mainly via an isopeptide bond between the Ub C-terminal glycine carboxylic group and lysine ε-amino group). Protein ubiquitination is a sequential enzymatic cascade reaction, in which Ub is sequentially added to the protein substrate using Ub-activating enzyme (E1), conjugating enzyme (E2), and ligase (E3) [4]. The ubiquitinated proteins are then degraded by the 26s proteasome, while the Ub is removed from the substrate by deubiquitinating enzymes (DUBs) and enters the next ubiquitination pathway. This system constitutes the ubiquitin–proteasome system (UPS). UPS is the major protein hydrolysis control system in cells, which regulates the clearance of misfolded and aggregated proteins and most excess produced normal proteins [17].

Ubls are a class of PTM proteins structurally similar to Ub. Most of them have a similar β-grasp fold and a conserved C-terminus with one or two glycine motifs that conjugate to substrates [18,19]. Dozens of Ubls, such as small ubiquitin-like modifiers (SUMOs), neural precursor cell-expressed developmentally downregulated gene 8 (NEDD8, also known as Rub1), autophagy-related protein 8 (ATG8), autophagy-related protein 12 (ATG12), ubiquitin-related modifier 1 (URM1), ubiquitin-fold modifier 1 (UFM1), HLA-F-adjacent transcript 10 (FAT10), and UBL3, have been identified as PTM proteins. The number of substrates recognized by different Ubls varies widely, with Ub recognizing millions of substrates [20]. These Ubls have been shown to be involved in oncogenic and tumor-suppressor pathways in many types of cancers, while some cases of Ubl modifications are not directly involved in proteolysis.

## 3. Ubls and Cancers

### 3.1. SUMO

PTMs with SUMOs are involved in various cellular processes, including cell-cycle progression and DNA damage response [21]. The binding of SUMO to substrate proteins is called SUMOylatoin [22] and three subtypes of SUMOs are known as SUMO1, SUMO2, and SUMO3. SUMOylation involves a range of cascade proteins, including SUMO E1; SUMO-activating enzyme subunits (SAE1/SAE2 complex), SUMO E2; Ubiquitin-conjugating enzyme E2 I (UBC9), SUMO E3; a variety of types, such as protein inhibitor of activated STAT family [23]; and SUMO/sentrin-specific peptidases (SENPs). The expression of those proteins is known to be involved in various cancers (Table 1). At present, SAE1/SAE2, UBC9, and SENPs are the targets of cancer therapeutics. Inhibitors of SAE1/SAE2 and UBC9 block the SUMOylation cascade, while inhibitors of SENPs block deSUMOylation of a subset of their targets. To date, no small-molecule inhibitors of SUMO E3 have been identified. For example, TAK-98, an inhibitor of SUMO E1 synthesis 1, is the only SUMOylation inhibitor currently being evaluated in three cancer clinical trials (i, ii, iii). Spectomycin B1, also known as an antibiotic, binds to UBC9 and inhibits the formation of E2-SUMO intermediates, inhibiting the estrogen-dependent growth of human breast cancer cells [24].

### 3.2. NEDD8

Neuronal precursor cell-expressed developmentally downregulated protein 8 (NEDD8) is encoded by the *NEDD8* gene in humans [2]. NEDD8 is a well-known Ubl, which has the highest similarity of 76% in amino acid composition with Ub [25]. NEDD8 functions in embryogenesis and controlling the cell cycle by conjugating to some cellular proteins via the process of NEDDylation. Cullin-group proteins are well-known substrates of NEDD8 [26]. The molecular scaffold for Cullin–RING ligases (CRLs) is provided by Cullins [27,28]. In renal carcinoma cells, NEDDylation of von Hippel-Lindau protein was downregulated [29]. NEDDylation over-activation in tumor cells controls tumor-derived signals to enhance the tumor microenvironment [30]. Downregulation of NEDD8 was found in prostate malignancy [31]. NEDDylation was enhanced in squamous cell carcinoma cell lines [32]. The therapeutic strategy targeting the NEDD8 pathway is currently being investigated. Recently, MLN4924, an inhibitory drug of the NEDD8-activating enzyme, was found to be potent in acute myeloid leukemia, melanoma, and solid [33].

### 3.3. ISG15

The modification with the Interferon-stimulated gene (ISG15) is known as ISGylation. ISG15 was the first identified Ubl [19,34]. ISG15 is overexpressed in pancreatic ductal adenocarcinoma, where tumor-associated macrophages secrete ISG15 [35]. Ubiquitin-activating enzyme E1-like protein, an enzyme responsible for ISGylation, is downregulated in lung cancer cells [36]. Constitutive expression of ISG15 is associated with melanoma by inciting E-Cadherin expression on dendritic cells in human [37]. ISG15 is overexpressed in triple-negative breast cancers and contributes to its progression [38]. Similarly, ISGylation level was found to be higher in hepatocellular carcinoma and colon cancer compared to normal tissues [39,40]. In nasopharyngeal carcinoma cells, the expression of ISG15 upregulated the cancer stem cell phenotypes, such as pluripotency, tumorigenicity, and tumor-sphere formation [41]. Currently, few drugs (i.e., irinotecan and topotecan) targeting the ISG15 pathway are available to treat colorectal and ovarian cancers [42].

### 3.4. ATG8

ATG8 includes two subfamilies, the gamma-aminobutyric acid type A receptor-associated protein (GABARAP) and microtubule-associated protein 1 light-chain 3 (LC3) subfamilies, both of which are involved in the process of autophagy induction to autophagosome–lysosome fusion via conjugation to the lipid phosphatidylethanolamine (PE) and have been elucidated to play an important role in cancer progression [43]. The levels of autophagic flux in cancer cells directly affect the expression levels of ATG8 family proteins [44]. ATG8 family proteins apparently act as tumor suppressors, mainly by participating in the selective degradation of oncogenic proteins or damaged organelles [45]. On the other hand, some researchers also reported that the expression of LC3 was upregulated and facilitates the progression of patients with gastrointestinal cancers, especially in early carcinogenesis [46]. The change in ATG8 expression levels was positively correlated with a good prognosis in several cancer types, including hematopoietic cancer, renal cancer, prostate cancer, breast cancer, hepatocellular carcinoma, and gastric cancer [47,48,49,50,51,52,53]. In summary, this evidence supports that ATG8 could be a potential therapeutic target to develop new anticancer therapies.

### 3.5. ATG12

Another autophagy protein, ATG12, participates in the formation and expansion of autophagosomal membranes by forming an irreversible ATG12–ATG5–ATG16L1 complex together with ATG5 and ATG16L1. This series of cascade mechanisms is similar to the E1-E2-E3 cascade in the ubiquitination pathway [54]. Single-nucleotide repeat shift mutations in *ATG12* may be involved in the carcinogenesis and progression of gastric and colorectal cancer through the deregulation of autophagy [55]. However, the mechanism of ATG12 expression in tumorigenesis is still unclear. In head and neck squamous cell carcinoma, lower ATG12 expression was correlated with better prognosis [56]. On the other hand, drug or gene therapy inhibiting ATG12 has been reported to improve the prognosis of gastric cancer, breast cancer, renal cancer, and pancreatic cancer [57]. Hu J L et al. also reported that inhibition of ATG12-mediated autophagy sensitized colorectal cancer cells to radiation therapy [58]. Therefore, ATG12 is worthy of further investigation as a potential anticancer therapeutic target.

### 3.6. FAT10

The FAT10 protein is known as human leukocyte antigen-F adjacent transcript 10 or ubiquitin D (UBD), which was first identified by Fan et al. in 1996 [59]. The main role of FAT10 modification is to mediate proteasomal degradation of the target protein [60]. FAT10 is also involved in cancer development. For example, FAT10 upregulated HOXB9 in hepatocellular carcinoma (HCC), through the β-catenin/TCF4 signaling pathway and facilitates HCC invasion in vitro [61]. In bladder cancer, FAT10 promoted the proliferation of cancer cells through direct interaction with Survivin [62]. Moreover, FAT10 expression was highly upregulated in many different cancer types, such as colon and breast cancer, where it enhanced cancer cell migration, invasion, and metastasis formation [63]. In addition, FAT10 is known to regulate several pathways, such as NF-κB, Akt, or Wnt signaling, involved in cancer development, and also to directly interact with downstream targets, such as p53, β-catenin, SMAD2, and MAD2, leading to enhanced survival, proliferation, invasion, and metastasis formation of cancer cells but also of non-malignant cells [64]. As FAT10 overexpression has significant pro-malignant properties in different cancer types, and FAT10 affects substrate proteins and results in bio-functional change, it could be a novel therapeutic strategy based on targeting FAT10 signaling for cancer treatments [65].

## 4. UBL3

UBL3 was originally identified as a novel Ubl in *Drosophila melanogaster* in 1999 [66]. Its orthologues were later revealed to be present also in animals, filamentous fungi, and plants, and reported to be a membrane-anchored UB-fold protein [5]. After the human and mouse orthologues were cloned and sequenced, *UBL3* was found to be located on human chromosome 13q12-13 and mouse chromosome 5 telomeres (MMU5), within a region of gene order shared by human and mouse chromosomes. Although it was not found in yeast, UBL3 is considered to be a highly conserved Ubl like Ub, SUMO, and NEDD8. Although UBL3 has been identified in a variety of species for decades and its expression is widespread in different human organs, mainly including the brain, lung, kidney, stomach, esophagus, colon, endometrium, gall bladder, and thyroid, the role of UBL3 has been less studied.

### 4.1. UBL3 Structure

UBL3 has structural homology with Ub [5]. UBL3 in humans has 117 amino acids, of which residues 10 to 88 are Ub-folding structural domains. Like all proteins in the β-grasp structural domain superfamily, UBL3 consists of five β strands and an α helix. The β fold forms a mixed β fold in the order 2-1-5-3-4, and the α helix (residues 31–41) is located in the notch of the β fold. This is similar to other Ubl structural domains, including SUMOs, NEDD8, ATG8, ATG12, etc. Another feature of UBL3 is the carboxy-terminal CAAX box (C is a cysteine, A is an aliphatic amino acid, X is any amino acid) replacing the C terminus ending in a di-glycine motif in Ubl. The CAAX box is a typical motif for protein isoprenylation. The CAAX sequence of UBL3 is composed of C, valine (V), isoleucine (I), and leucine (L). Its 113th amino acid cysteine, linked to this CAAX box, forms the end of two consecutive cysteine residues, CCVIL. In the research of Ageta et al., the mutation in residue C114, a part of the CAAX motif, reduced the membrane localization of UBL3 compared to mutations in C113 [7]. Therefore, C114 undergoes isoprenylation and is involved in membrane anchoring of UBL3. This is consistent with the findings of Downes et al. that the C114 residue of UBL3 in AT is responsible for membrane localization [5].

### 4.2. UBL3 Function

The role of UBL3 as a post-transcriptional modifier (Figure 1) was revealed [7]. The protein modification with UBL3 is completely different from the modification of Ub and other Ubls. The two consecutive glycine residues at the C terminus in Ub, SUMOs, and Nedd8 are involved in the modification process by conjugating to the lysine residues of the target proteins through an isopeptide bond. In the case of UBL3, the characteristic terminus, CCAAX, is thought to be the key for UBL3 to exert as a PTM factor [7]. The mutation in the CCAAX sequence reduced the UBL3 modification levels in the MDA-MB-231 cells. However, details on the chemical modification between UBL3 and a target protein are not yet known. In *Ubl3*-knockout mouse, the amount of sEV proteins was reduced. A comprehensive proteomic analysis by Ageta et al. identified 1241 proteins interacting with UBL3, 29% of which were annotated as “extracellular vesicle exosomes” from Gene Ontology (GO) for enrichment analysis. These suggest that UBL3 modification is involved in the sorting of exosome proteins. Among those UBL3-interacting proteins, a variety of cancer-related proteins, such as HRAS, KRAS, TGFBR1, RB1, ITGA6, ITGB4, mTOR, TSC2, and APLP2, is included. These imply that UBL3 may play an important role in cancer invasion and metastasis via sEVs. sEVs are mediators of intercellular communication, and sEV-mediated cancer-normal cell communication in the tumor microenvironment is a key process in cancer progression [67]. Cancer-derived sEVs promoted the spread and colonization of cancer cells by remodeling the tumor microenvironment and altering the extracellular matrix [68]. Cancer sEVs play a role in six major areas: angiogenesis, epithelial mesenchymal transition (EMT), invasive metastasis, immune escape, cancer-associated fibroblasts (CAFs), and drug resistance [69]. UBL3 may influence these processes by being involved in sorting proteins into sEVs.

UBL3 has also recently been identified as a key regulator of MARCH-mediated ubiquitination of MHC II and CD86 [12]. C114, which is an isoprenylation site and located at the C-terminus CAAX motif of the UBL3 protein, was a key site for UBL3 to participate in MARCH-mediated MHC II ubiquitination, suggesting that membrane anchoring of UBL3 is a prerequisite for participation in this process [6,12]. UBL3 may regulate the ubiquitination of MHC II by transferring an E2 enzyme to the cell membrane anchor site of MARCH1, a Ub-E3 ligase (Figure 1). MHC II is an antigen-presenting molecule that is mainly expressed on antigen-presenting cells (APCs), such as dendritic cells (DCs), macrophages, and B cells [70]. MARCH-mediated ubiquitination of MHC II is essential for its maintenance of endocytosis, recirculation, and turnover rate [71,72]. Defective ubiquitination of MHC II caused functional and immunological defects in dendritic cells, such as reduced number of splenic DCs, altered DC cytokine production, impaired hydrolysis of Ag proteins in DCs, and, ultimately, reduced antigen presentation in vivo, impaired cytotoxic T lymphocytes immune response, and suppressed humoral immunity [73]. Similar functional defects in DCs were observed in Ubl3-knockout mice [11]. By presenting antigens, DC-mediated T-cell activation can elicit antitumor immune responses, which are essential for antitumor immune responses [74]. It has been found that DC infiltration is positively correlated with T-cell infiltration and cancer survival, and that preventing DC cell infiltration into the tumor microenvironment is one of the strategies for tumor escape [75]. Therefore, DCs are widely explored as tools for immunotherapy. UBL3, as a key molecule affecting DC function, must have the same potential in immunotherapy of cancer. Further studies are needed to determine whether tumorigenesis mediated by immunodeficiency is related to UBL3 and whether UBL3 overexpression can improve the body’s immune response to tumors.

## 5. UBL3 and Cancers

### 5.1. Lung Cancer

Lung cancer is the most-common cause of cancer-related deaths, causing 1.8 million deaths worldwide each year [76]. Lung cancer includes small-cell lung cancer (SCLC) and non-small-cell lung cancer (NSCLC), of which NSCLC accounts for 80% of lung cancers in the United States and two-thirds is diagnosed at advanced stages [77]. Zhao et al. found that the expression of UBL3 was significantly downregulated in non-small-cell lung cancer and the tumor growth was reduced in the xenograft of A549 NSCLS cells expressing UBL3 to mice compared to the parent A549 cells [78]. It has also been shown that the expression of UBL3 positively correlated with NSCLC patient survival. This result is consistent with a positive correlation between DC and progression-free survival of NSCLC patients [79], from the view point that UBL3 affects DC function. This gives rise to the speculation that UBL3 may maintain dendritic cell function and immunity by mediating MARCH ubiquitination of MHC II and stimulating the body’s immune response to tumor cells. Another interesting finding was that UBL3 expression was higher in non-smoking NSCLC patients than in NSCLC patients with a history of smoking [78]. It has also been observed in mouse and human DC cell lines that smoking extracts impeded the ability of DCs to stimulate and activate antigen-specific T cells [80,81]. Whether UBL3 is involved in the process of smoking leading to impaired antigen presentation in DC cells is an intriguing direction to investigate.

UBL3 itself and some sEV proteins are known to be related to NSCLC (Table 2). For example, epidermal growth factor receptor (EGFR) was overexpressed in serum sEVs of NSCLC patients, and the EGFR protein level of sEVs was negatively correlated with overall survival [82]. The sEV protein transforming growth factor beta (TGF-β) regulated invasion and metastasis through the loss of epithelial markers and the acquisition of mesenchymal markers in lung cancer cell lines. Its induction of EMT is the main feature of invasion and metastasis in tumor progression [83,84]. TGF-β induced demethylation of H3K27Me3 in the *SNAI1* promoter and overexpression of SNAI1 in lung cancer cells leading to EMT [85].

### 5.2. Breast Cancer

Breast cancer (BC) surpassed lung cancer in women as the most-common cancer and fifth-leading cause of cancer death worldwide in 2020 [76]. Lee et al. identified UBL3 as a novel susceptibility locus using a genome-wide association study (GWAS) for Asian women carrying BRCA1/2-Negative, high-risk breast cancer (BRCAX) [111]. Ephrin type A receptor 2 (EhpA2) is one of the UBL3-interacting proteins [7]. Gao et al. found that the sEV EphA2 from drug-resistant cancer cells activated the ERK1/2 signaling pathway by inducing reverse Ephrin signaling to stimulate aggressive phenotypic metastasis of drug-sensitive cancer cells and promote migration and invasion of breast cancer [89]. UBL3 also interacted with integrins and integrin-related proteases [7]. α6β1 integrin was detected in breast cancer and the expression of the α6β1 integrin was significantly increased in lymph node metastases, indicating that the α6β1 integrin plays an important role in the migration of breast cancer [112]. The α3β1 integrin promoted the production of prostaglandin E2, which, in turn, induces breast cancer tumor angiogenesis and metastasis [113]. Lin et al. found that ADAMs and MMPs, both of which are integrin-related proteases, were upregulated by aspartate β-hydroxylase in the process of invasive and metastatic breast cancer cells [91]. This upregulation was mediated by sEVs, specifically containing Notch receptors with ligands [91]. The expression of ADAM10 in CD9-positive sEVs increased in breast and ovarian cancers in another study [114]. In summary, UBL3 may play an important function in the process of invasion and metastasis of breast cancer, but further studies are needed to verify this.

### 5.3. Gastric Cancer

Gastric cancer (GC) is a common cancer of the digestive tract, with approximately 1 million new cases of gastric cancer pathology worldwide in 2020, accounting for 5.6% of all cancers [76]. One article reported that UBL3 was significantly downregulated in gastric cancer by RNA-seq measuring 24 pairs of cancerous tissues versus normal gastric tissues [115]. Among the UBL3-interacting proteins, intercellular adhesion molecule (ICAM)-1, proteasome 20S subunit α3 (PSMA3), PSMA6, CD97, and CD44 were reported to be involved in the progression of GC through the sEV system. ICAM-1, secreted from the omental tissue, increased the migratory and invasive properties of gastric cancer cells in vitro [92]. PSMA3 and PSMA6 were found to be remarkably higher in serum sEVs of patients with metastatic gastric cancer (mGC) than in healthy controls and patients with early GC [93]. This suggests that proteasome subunits may be involved in GC metastasis via sEVs. CD97, sEV-dependent, from high lymph node metastatic GC cells promoted self-lymph node metastasis and play an essential role in the formation of pre-metastatic ecological sites [94]. sEV CD44 from GC cells activated YAP-CPT1A-mediated (yes-associated protein and carnitine palmitoyltransferase 1A) reprogramming of fatty acid oxidation to deliver lymph node metastatic capacity between GC cells [95]. In summary, UBL3 may act as a promoter of tumor invasion and metastasis in GC, while there is inconsistency about the apparent downregulation of UBL3 in GC. It may be that UBL3 plays different roles in different stages of cancer.

### 5.4. Pancreatic Cancer

Pancreatic cancer (PC) is one of the most malignant tumors with a very poor prognosis and it has the lowest five-year survival rate of all tumors in the United States at 11% [116]. Liu et al. constructed a three-gene risk profile containing UBL3, CDKN2A, and BRCA1for predicting overall survival in pancreatic cancer patients [117]. The study used multivariate cox regression analysis to show that three-gene characteristics could be an independent predictor of prognosis in pancreatic cancer patients. Among them, UBL3 expression was significantly downregulated in the high-risk group and negatively correlated with risk score [117]. EphA2 and MMP14, which are UBL3-interacting sEV proteins, are known to be key molecules for PC drug responses. EphA2 can transfer chemo resistance from Gemcitabine (GEM)-resistant PC cells to GEM-sensitive PC cells via sEVs [99]. sEV-mediated transfer of MMP14 enhanced drug resistance and promoted sphere formation and migratory capacity of receptor-sensitive pancreatic ductal adenocarcinoma cells [100]. In the context of MHC II, UBL3 may also be involved in the immune system in PC. sEVs enriched with tumor-associated antigens, such as tubulin beta (TUBB), further depleted antibodies with associated immune cells by binding to circulating autoantibodies, leading to the suppression of complement-dependent cytotoxicity and antibody-dependent cell-mediated cytotoxicity [118]. In pancreatic ductal adenocarcinoma, antigen-presenting CAF showed a high expression of MHC II and appears to play an important immune role in the immune microenvironment of this tumor [119]. Whether UBL3 can affect the expression of MHC II on the surface of this antigen-presenting CAF is worth exploring.

### 5.5. Other Cancers

The association of UBL3 with several other cancers has also been provided. Promoter methylation and downregulation of the *UBL3* gene were found in patients with esophageal cancer in northeast India [120]. LNCap showed upregulation of UBL3 expression after exposure to silvestrol, a natural product from the flavagline family [121]. A study of melanocytic tumors identified that UBL3 appeared as a fusion partner of MAP3K8 [122]. In patients with early-stage cervical cancer, UBL3 could be used to predict recurrence and survival with a negative correlation with relapse-free survival [123].

Considering the UBL3-interacting proteins, bladder cancer, and liver fibrosis, the latter of which is a major risk factor for hepatocellular carcinoma, may also be related to UBL3. In bladder cancer, EMT-mediated invasion occurred through TGF-β in sEVs. Hepatic stellate cells (HSCs) secreted sEVs containing glycolysis-related molecules glucose transporter protein GLUT1 and pyruvate kinase M2 upon activation that are closely associated with liver fibrosis [124,125]. As for immune cells and UBL3, in tumor tissues of patients with esophageal cancer, low expression and function of tumor-infiltrating DCs have been reported, and the proportion of MHC-II-positive DCs was also significantly reduced, while it is not still clear whether the effect of UBL3 on DC cell expression and function via MHC II ubiquitination plays a role in tumors [126]. Overall, there is a relationship between UBL3 and the development, metastasis, and prognosis of some cancers, but more in-depth studies are still needed.

## 6. Prospective

UBL3 is downregulated in non-small-cell lung cancer, gastric cancer, esophageal cancer, and pancreatic cancer and it has been identified as a tumor suppressor in NSCLC, while in breast cancer, it is a susceptibility gene for high-risk breast cancer and is an MAP3K8 fusion partner in melanocytic tumors. These reports seem that the UBL3 functions may be different in different cancers and at different stages of cancer. As we mentioned, regarding the hypothesis in [127], there are two possible functions played by UBL3 in cancers: UBL3 acts as a “guardian” in cells to inhibit cancer development and metastasis by collecting tumor-promoting-related factors in cancer cells and isolating these factors into sEVs. For example, MMP14 is involved in angiogenesis, extracellular matrixization, and epithelial mesenchymalization in breast cancer, pancreatic cancer, colon cancer, and sarcoma, which promote tumor invasion and metastasis [128,129,130,131]. The interaction of UBL3 with MMP14 may be to segregate MMP14 into sEVs to inhibit cancer progression. For the alternative hypothesis, UBL3 is involved in the dissemination of tumor factors mediated through sEVs, promoting tumor metastasis, invasion, and drug resistance. Two proto-oncogene proteins, KRAS and HRAS, are also identified as UBL3-interacting proteins, and RasG12V, a constitutively active one, caused its downstream activation via sEVs isolated from MDA-MB-231 cells depending on UBL3 [7]. Whether UBL3 is an inhibitor or a promoter of cancer, UBL3 has the potential to be a target for cancer therapy. Alterations in sEV-related content in cancer cells by inhibiting or activating UBL3 function inhibits cancer cell invasion and metastasis.

There is also potential for UBL3 function in cancers that tumor tissues lead to defective MHC II ubiquitination through downregulation of UBL3, thus, affecting DC cell function as one approach for tumor cell immune escape. This seems to explain the downregulation of UBL3 and the correlation between UBL3 expression and survival in lung cancer. A highly immunosuppressed tumor microenvironment, characterized by reduced numbers and impaired function of effector T cells and antigen-presenting cells, especially dendritic cells, remains a key obstacle to improving the efficacy of tumor immunotherapy [132,133]. If UBL3 can cause infiltration of DCs CD4+ and CD8+ in tumors, which, in turn, enhances the body’s immune response to tumor cells, this would address the problem of tumor immunosuppression from another perspective.

There is widespread interest in using sEVs as a tool for treating a range of diseases, including cancer [134,135]. sEVs, as therapeutic tools, have advantages, such as easier blood circulation clearance, large capacity for ex vivo expansion, biocompatibility due to their endogenous origin, and high cellular uptake [135,136,137]. In particular, sEVs can overcome biological barriers, such as the blood–brain barrier and lung clearance [138,139]. UBL3, a key molecule for sorting sEV proteins, can be used as a tool for engineering the desired proteins and for generating sEVs.

## 7. Conclusions

Using UBL3, an emerging Ubl, as a targeting site to regulate sEV protein and immune function in cancer will provide a potential new direction for cancer treatment. UBL3 has been found to play a critical role in sEV protein sorting and maintenance of the body’s immune function. The relationship between UBL3 and various cancers has also been gradually uncovered. However, further studies on the function of UBL3, its mechanism of action, and its effects on various cancers are still needed to understand UBL3 more deeply.

## Figures and Tables

**Figure 1 ijms-24-01231-f001:**
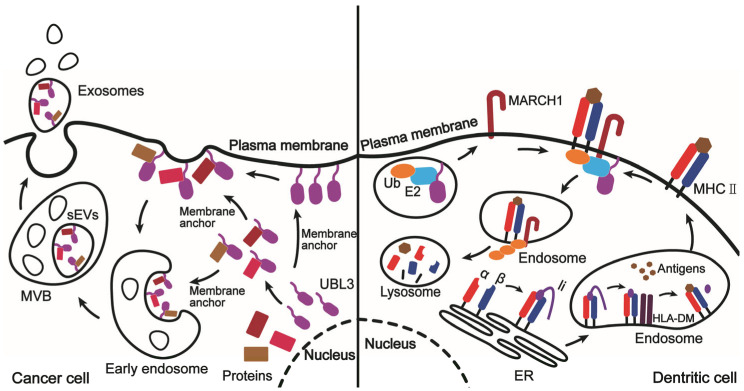
Model diagram of Ubiquitin-like protein 3 (UBL3) function. Left part: Proteins are modified with UBL3, which is free or membrane-anchored (exact mechanism of modification is not clear). The modified proteins are recruited into the endosomes, forming multivesicular bodies (MVBs) containing small extracellular vesicles (sEVs), and sEVs are secreted out of the cell. Right part: major histocompatibility complex class II (MHC II) α and β chains are assembled and complexed with Ii in the endoplasmic reticulum (ER). Upon entry into the endosome, Ii is degraded to class II-associated invariant chain peptide (CLIP). HLA-DM catalyzes the release of CLIP and binding of antigen. The stable antigen–MHC II complex is transferred to the cell surface to present the antigen to T cells. UBL3 recruits a Ub-E2 enzyme to the membrane and promotes ubiquitination of MHC II by membrane-associated RING-CH (MARCH) 1, a Ub-E3 ligase, and then MHC II is targeted into the lysosome for degradation.

**Table 1 ijms-24-01231-t001:** Identity with Ub, representative substrates, and related cancers of Ubls.

Ubls	Identity with Ub	Representative Substrates	Related Cancers
SUMO	18% *	p53, c-JUN, IκBα	Acute promyelocytic leukemia, Acute myeloid leukemia, lymphoma, Myeloma, Glioma, Hepatocellular carcinoma, Lung cancer, Breast cancer, Prostate cancer
NEDD8	76%	Cullins, pVHL, p53	Prostate cancer, Renal cancer, Squamous cell carcinoma, Acute myeloid leukemia, Melanoma
ISG15	31%	Cyclin D1, p63, HIF-1α	Pancreatic ductal adenocarcinoma, Lung cancer, Melanoma, Triple-negative breast cancer, Breast cancer, Hepatocellular carcinoma, Colon cancer, Nasopharyngeal carcinoma, Ovarian cancer
ATG8	14%	NBR1, NDP52, Rab7	Hematopoietic cancer, Renal cancer, Prostate cancer, Breast cancer, Hepatocellular carcinoma, Gastric cancer, Head and neck carcinoma, Thyroid cancer, Colorectal cancer, Esophageal squamous cell carcinoma, Melanoma
ATG12	10%	ATG3, Bcl-2	Gastric cancer, Colorectal cancer, Head and neck squamous cell carcinoma, Breast cancer, Renal cancer, Pancreatic cancer
FAT10	36%	S5a, Survivin, p53	Hepatocellular carcinoma, Non-small-cell lung cancer, Gastric cancer, Breast cancer, Bladder cancer, Colorectal cancer, Cervical cancer, Glioma

*: SUMO1.

**Table 2 ijms-24-01231-t002:** Summary of the relationship between UBL3 and tumors.

Cancer	UBL3 Expression	Effect of UBL3 on Survival/Risk	Tumor-Related sEV Proteins Interacting with UBL3
Lung cancer	Downregulation *	Protective *	TGF-β, Induce epithelial mesenchymal transition in BALB/c nude mice [85] EGFR, Inhibition of tumor antigen-specific CD8+ cells in vitro [86] KRAS, Induced CD4+ T phenotypic conversion to Treg-like cells that are immune-suppressive in vitro [87]
Breast cancer	Upregulation ^#^	High risk ^#^	TGF-β, Increased myofibroblast-like phenotype of adipose tissue-derived mesenchymal stem cells in vitro [88] EphA2, Activation of EPK1/2 signaling promotes cancer metastasis in xenograft tumor model SCID mice [89] Integrin α6, Promotion of lung metastasis of breast cancer cells in NCr nude mice [90] ADAM, MMP, Promotion of breast cancer metastasis in BALB/c nude mice [91]
Gastric cancer	Downregulation	Unclear	ICAM-1, Promotion of AGS gastric cancer cells metastasis in vitro [92] PSMA2, PSMA6, Upregulated in metastatic gastric cancer patient exosomes [93] CD97, CD44, Promotion of lymph node metastasis in BALB/c nude mice [94,95] MET, Mediated the pro-tumorigenic effects of macrophages in nude mice [96] EGRF, Promote gastric cancer liver metastasis in BALB/c nude mice [97] ITGβ5, Promotes N87 gastric cancer cells migration and invasion in vitro [98]
Pancreatic cancer	Downregulation	Protective	EphA2, Promotes transfer of drug resistance of pancreatic cancer cell lines [99] MMP14, Promotion of pancreatic ductal carcinoma cell migration [100] CD44, Enhance PC cells migration and invasion in SCID mice [101] CKAP4, Proliferation and migration of pancreatic ductal adenocarcinoma cells [102]
Esophageal cancer	Downregulation	Unclear	O-GlcNAc, Promote the immune escape of ALDH^+^ cells [103]
Prostate cancer	Unclear (Upregulated after exposure to silvestrol)	Unclear	PKM, Promotion of bone metastasis in SCID mice [104] ITGA3, Promotes epithelial cell migration in vitro [105] MMP14, Promotion of prostate cancer cell growth in vitro [106] Rab1A, Rab1B, Rab11A, Promotion of prostate cancer cell growth in nude mice [107] HSP90, ILK, Increase stemness, metastasis, and CAFs formation in vitro [108]
Melanocytoma	Unclear (Involved in MAP3K8 gene fusion)	High risk	Rab1A, Rab5B, RAB27A, Induced vascular leakiness at pre-metastatic sites in C57BI/6 mice [109] Met, Enhance lung metastasis in C57BI/6 mice [109]
Cervical cancer	Unclear	High risk	TGF-β, Induce T regulatory cell expansion in vitro [110]

* in NSCLC; ^#^ in BRCAX.

## Data Availability

Not applicable.

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
