# Peer review of "A New Potential Therapeutic Target for Cancer in Ubiquitin-Like Proteins—UBL3"

_ijms, 2023, doi:10.3390/ijms24021231_

Round 1
Reviewer 1 Report
Hengsen Zhang et. al., presented a comprehensive review describing ubiquitin-like proteins that is very interesting indeed. I particularly enjoy the narrative starting from the plant world, important basis of biology many times forgotten or oversighted. The authors elegantly amalgamated a substantial collection of up-to-date references and constructed an excellent summarising table. The review would be a must-read for anyone who like to be introduce to the Ubiquitin-like proteins field and I consider it is a great asset for the journal. I would recommend it promptly publication.
Comments:
At table 1: Replace note labelling by symbols (e.g., * § #), therefore will not be confused with reference numbers.
At figure 1: Schematic diagram is very informative and simple to follow. However, would largely improve with bigger and bold labelling, ticker lines, and bigger draws. Black and dark blue are difficult to distinguish each other.
Author Response
Responds to the reviewer’s comments:
Comment 1: At table 1: Replace note labelling by symbols (e.g., * § #), therefore will not be confused with reference numbers.
Response: We have used * and # to replace the note labeling in table 2. (In line272 , Page 8)
Comment 2. At figure 1: Schematic diagram is very informative and simple to follow. However, would largely improve with bigger and bold labelling, ticker lines, and bigger draws. Black and dark blue are difficult to distinguish each other
Response: We are so grateful for your kind suggestion. We have adjusted the labeling, ticker, lines, and draws in Figure 1 as suggested by the reviewer. And the black marked the “Antigen” have been changed to brown. (In line 211, Page 5)
Reviewer 2 Report
General notes:
-
I think the paper is structurally well organized. However, as of now, it is just large blocks of text, with just one figure, one table, and a flurry of references. So it is not easy for the eye to read and discuss. I think some of the points made here can be visualized. For example, the authors have listed many PTMs, like SUMO and NEDD4, but again, just blocks of text.
-
The Paper is written in great detail. Maybe even too many details.
-
The reference volume is really high, maybe more than necessary. There are 195 of them. This feels like a long review paper to me.
-
While there is only one figure and one table, they are well prepared. Figure legend explains the figure well and the table helps understand the Ubl3-Cancer relationship.
-
I know this is not an experimental paper, but I feel like there needs to be more data on how Ubl3 is a good target. I feel like there is too much information, some irrelevant to the topic of the Title.
-
All in all, this reads a lot like a book chapter about Post Translational Modifications.
Line-by-line detailed notes:
-
Starting from line 60, the authors start to mention CD68 and other CDs as in line 62 CD4+, CD8+; but they could write the long version of the abbreviation at least once throughout the article.
-
The sentence in line 68: “Ub is a highly conserved protein, consisting of 76 amino acids, and is commonly expressed in all eukaryotic cells [17].” Although this sentence is a general statement, the 17th reference does not mention that protein is expressed in all eukaryotic cells. The following article can be a more accurate reference: https://www.ncbi.nlm.nih.gov/pmc/articles/PMC8122580/ Also, the words “commonly” and “all” do not complement each other.
-
A clear explanation for the subfamilies of ATG8 is provided in line 163, but they continue to mention LC3 throughout the paragraph. The long version of this abbreviation could also be provided.
-
The following sentence in line 187 “Although the mechanism of ATG12 expression in tumorigenesis is still unclear.” seems to be missing. The word “Although” can be substituted with another conjunction to prevent this wrong assumption.
-
Since there is already a plethora of information throughout the article, also the examples can be mentioned specifically rather than et al: “in different murine organs, including the brain, small intestine, lung, spleen et al,...” in line 227.
-
There are some grammatical errors throughout the article:
-
Line 98: “SUMOylation have a range of cascade proteins involved,...” Instead of “have”, “has” must be used.
-
Line 148: “... where tumor-associated macrophages secretes the ISG15…” Instead of “secretes”, the usage must be “secrete”.
-
Line 154: “... level of ISGylated proteins were higher…” “Was” should have been used.
-
Line 199: “... It is mainly express in organs…” This sentence started as a passive voice and continued as an active voice.
Line 261: “... involved in the modification process by conjugate to the lysine residues…” The gerund of the verb conjugate should be used.
Author Response
Responds to the reviewer’s comments:
General notes:
Comment 1: I think the paper is structurally well organized. However, as of now, it is just large blocks of text, with just one figure, one table, and a flurry of references. So it is not easy for the eye to read and discuss. I think some of the points made here can be visualized. For example, the authors have listed many PTMs, like SUMO and NEDD4, but again, just blocks of text.
Response: Following your suggestion, we have summarized the information on PTMs mentioned in the first half of the review and created a table. (Table 1. Identity with Ub, representative substrates, and related cancers of Ubls. Line 106, Page 3)
Comment 2: The Paper is written in great detail. Maybe even too many details.
Response: Thank you for this valuable feedback.
We have made appropriate adjustments and streamlined the details in the “Ubls and cancers” sections.
In 3.1. SUMO, we deleted the second paragraph describing drugs targeting SUMO for the treatment of cancer and provided a brief example at the end of the first paragraph. (Line 101-105, Page 2)
In 3.3. ISG15, we have adjusted the description of the relationship between ISG15 and cancer by removing the description of "UBP43 as de-ISGylation enzyme for cancer treatment". (Line 128, Page 3); The example of "ISGylated protein changes in breast cancer-derived tumor cells" has been removed. (Line 129, Page 3); Integrated the description of "ISGylation changes in hepatocellular carcinoma and colon cancer". (Line 130, Page 3)
In 3.4. ATG8, we removed the overly detailed examples of cancers associated with ATG8. (Line 149, Page 4)
In 3.6. FAT10, we removed the overly detailed description of FAT10 and streamlined several sentences. (Line168-170, Page 4)
Comment 3: The reference volume is really high, maybe more than necessary. There are 195 of them. This feels like a long review paper to me.
Response: Thank you very much for your comment. When I adjusted and streamlined the details of the article, I also removed some relatively repetitive and less important references as appropriate. (From 195 to 139)
Comment 4: While there is only one figure and one table, they are well prepared. Figure legend explains the figure well and the table helps understand the Ubl3-Cancer relationship.
Comment 5: I know this is not an experimental paper, but I feel like there needs to be more data on how Ubl3 is a good target. I feel like there is too much information, some irrelevant to the topic of the Title.
Response: We are so grateful for your kind question.
According to comment 2, we removed some details. At the same time, the following areas that were not very relevant to the topic were also adjusted.
In 1. Introduction, we have removed the sentence that is not relevant to the topic. (Line41, Page 1)
In 2. Ub and Ubls, we have adjusted the description of Ub to remove details that are not relevant to the topic. (Line 77, Page 2)
In 4.2. UBL3 Function, we removed sentences with slightly unimportant details. (Line 231, Line242 and Line 260, Page 6)
Comment 6: All in all, this reads a lot like a book chapter about Post Translational Modifications.
Response: We have modified and adjusted the section of Post Translational Modifications to reduce its length in this review.
Line-by-line detailed notes:
Comment 7: Starting from line 60, the authors start to mention CD68 and other CDs as in line 62 CD4+, CD8+; but they could write the long version of the abbreviation at least once throughout the article.
Response: We have added the long version of the CD at the place where it first appeared. (Line 58, Page 2)
Comment 8: The sentence in line 68: “Ub is a highly conserved protein, consisting of 76 amino acids, and is commonly expressed in all eukaryotic cells [17].” Although this sentence is a general statement, the 17th reference does not mention that protein is expressed in all eukaryotic cells. The following article can be a more accurate reference: https://www.ncbi.nlm.nih.gov/pmc/articles/PMC8122580/ Also, the words “commonly” and “all” do not complement each other.
Response: Thank you very much for your suggestion. We have reconfirmed that the previous reference, as you said, did not mention that Ub is expressed in all eukaryotic cells. We have cited the reference you suggested. The “commonly” was replaced with “nearly”. (Line 67, Page2)
Comment 9: A clear explanation for the subfamilies of ATG8 is provided in line 163, but they continue to mention LC3 throughout the paragraph. The long version of this abbreviation could also be provided.
Response: Thank you very much for your suggestion. We have written the long version of the LC3 at the place where it first appeared. (Line 137, Page 4)
Comment 10: The following sentence in line 187 “Although the mechanism of ATG12 expression in tumorigenesis is still unclear.” seems to be missing. The word “Although” can be substituted with another conjunction to prevent this wrong assumption.
Response: Thank you very much for your suggestion. We have confirmed the logical relationship between the preceding and following sentences and replaced “although” with “but”. (Line 157, Page4)
Comment 11: Since there is already a plethora of information throughout the article, also the examples can be mentioned specifically rather than et al: “in different murine organs, including the brain, small intestine, lung, spleen et al,...” in line 227.
Response: We changed UBL3 in murine to in human and listed the organs that mainly express UBL3. (Line 191, Page5)
Comment 12: There are some grammatical errors throughout the article:
- Line 98: “SUMOylation have a range of cascade proteins involved,...” Instead of “have”, “has” must be used.
Response: We have corrected it. (Line 93, Page2)
- Line 148: “... where tumor-associated macrophages secretes the ISG15…” Instead of “secretes”, the usage must be “secrete”.
Response: We have corrected it. (Line 125, Page 3)
- Line 154: “... level of ISGylated proteins were higher…” “Was” should have been used.
Response: This sentence has been removed because we have adjusted and deleted the Ubls section. Thank you very much for the correction. (Line 130, Page 3)
- Line 199: “... It is mainly express in organs…” This sentence started as a passive voice and continued as an active voice.
Response: This sentence has been removed because we have adjusted and reduced the Ubls section. (Line 168, Page4)
- Line 261: “... involved in the modification process by conjugate to the lysine residues…” The gerund of the verb conjugate should be used.
Response: We have modified “conjugate” to “conjugating”. (Line 226, Page 6)
Round 2
Reviewer 2 Report
All the raised points are addressed. Accept.
Author Response
Thank you for your comments.